# Composite Films of HDPE with SiO_2_ and ZrO_2_ Nanoparticles: The Structure and Interfacial Effects

**DOI:** 10.3390/nano11102673

**Published:** 2021-10-11

**Authors:** Asif A. Nabiyev, Andrzej Olejniczak, Akhmed Kh. Islamov, Andrzej Pawlukojc, Oleksandr I. Ivankov, Maria Balasoiu, Alexander Zhigunov, Musa A. Nuriyev, Fovzi M. Guliyev, Dmytro V. Soloviov, Aidos K. Azhibekov, Alexander S. Doroshkevich, Olga Yu. Ivanshina, Alexander I. Kuklin

**Affiliations:** 1ANAS Institute of Radiation Problems, Baku AZ1143, Azerbaijan; musa_nuriev@mail.ru; 2Joint Institute for Nuclear Research, 141980 Dubna, Russia; aolejnic@jinr.ru (A.O.); akhmed.islamov@gmail.com (A.K.I.); andrzej@jinr.ru (A.P.); ivankov@jinr.ru (O.I.I.); masha.balasoiu@gmail.com (M.B.); dimkaupml@gmail.com (D.V.S.); azhibekoaidos@mail.ru (A.K.A.); doroh@jinr.ru (A.S.D.); ioyu@nf.jinr.ru (O.Y.I.); alexander.iw.kuklin@gmail.com (A.I.K.); 3Faculty of Chemistry, Nicolaus Copernicus University, 87-100 Torun, Poland; 4Institute of Nuclear Chemistry and Technology, 03-195 Warsaw, Poland; 5Institute for Safety Problems of Nuclear Power Plants NAS of Ukraine, 07270 Kiev, Ukraine; 6Moscow Institute of Physics and Technology, 141701 Dolgoprudny, Russia; 7Horia Hulubei National Institute of Physics and Nuclear Engineering, P.O. Box MG-6, RO-077125 Bucharest-Magurele, Romania; 8Institute of Macromolecular Chemistry, Czech Academy of Sciences, CZ-162 06 Praha, Czech Republic; zhigunov@imc.cas.cz; 9Faculty of Civil Engineering, Azerbaijan University of Architecture and Construction, Baku AZ1073, Azerbaijan; quliyevfevzi@mail.ru; 10Institute of Natural Science, Korkyt Ata Kyzylorda University, Kyzylorda 120001, Kazakhstan; 11The Institute of Nuclear Physics, Ministry of Energy, Almaty 050032, Kazakhstan; 12Donetsk Institute for Physics and Engineering Named after O.O. Galkin NAS of Ukraine, 03028 Kiev, Ukraine

**Keywords:** polymer-matrix nanocomposite, interface, fractal, lamellar thickness, SANS, SAXS

## Abstract

Herein, we investigated the influence of two types of nanoparticle fillers, i.e., amorphous SiO_2_ and crystalline ZrO_2_, on the structural properties of their nanocomposites with high-density polyethylene (HDPE). The composite films were prepared by melt-blending with a filler content that varied from 1% to 20% *v*/*v*. The composites were characterized by small- and wide-angle x-ray scattering (SAXS and WAXS), small-angle neutron scattering (SANS), Raman spectroscopy, differential scanning calorimetry (DSC), and scanning electron microscopy (SEM). For both fillers, the nanoaggregates were evenly distributed in the polymer matrix and their initial state in the powders determined their surface roughness and fractal character. In the case of the nano-ZrO_2_ filler, the lamellar thickness and crystallinity degree remain unchanged over a broad range of filler concentrations. SANS and SEM investigation showed poor interfacial adhesion and the presence of voids in the interfacial region. Temperature-programmed SANS investigations showed that at elevated temperatures, these voids become filled due to the flipping motions of polymer chains. The effect was accompanied by a partial aggregation of the filler. For nano-SiO_2_ filler, the lamellar thickness and the degree of crystallinity increased with increasing the filler loading. SAXS measurements show that the ordering of the lamellae is disrupted even at a filler content of only a few percent. SEM images confirmed good interfacial adhesion and integrity of the SiO_2_/HDPE composite. This markedly different impact of both fillers on the composite structure is discussed in terms of nanoparticle surface properties and their affinity to the HDPE matrix.

## 1. Introduction

The study of the structure and properties of polymer nanocomposites is one of the most important and interesting areas of materials science. This is because nano-sized fillers allow for reaching a high total volume in the filler/matrix interfacial region while preserving its relatively small thickness [1,2,3,4]. Hence, compared with traditional composites, it is possible to alter the properties of the material in different ways, even at low nano-filler concentrations.

Interfacial compatibility and adhesion are crucial for achieving the desired properties of composite materials [5,6]. Typically, high interfacial compatibility and adhesion are required since they allow researchers to obtain fatigue-resistant materials of high mechanical performance [7,8,9,10]. In particular, high interfacial compatibility and adhesion are key factors to achieve an effective transfer of the interfacial stress under mechanical load [7,9,11,12]. This in turn allows them to avoid undesired phenomena like the debonding of the filler particles from the matrix [7,13] and delamination in the vicinity of nanoparticles [12,14,15]. One special case is that of nanocomposites with weak interfacial adhesion [16]. These materials often have a closed-cell cellular structure, formed by interfacial voids and low-density regions. Since the presence of interfacial voids enables the accumulation of an electric charge, such composites are often used as piezoelectric materials [16]. Several studies have shown that interfacial adhesion depends not only on the mutual compatibility of the components (i.e., the matrix and the filler) but is also influenced by the composite preparation method and conditions. For instance, the composites prepared by melt blending with different quenching rates showed substantially different levels of interfacial adhesion and residual stress. From this standpoint, the level of interfacial integrity and adhesion appears to result from the interplay of various factors and requires further investigation.

Over the interfacial region, the polymer matrix can be perturbed; in particular, polymer chains can have different conformations, arrangements, packing types, and densities. The size of the lamellae and the degree of crystallinity can be influenced as well. The problem of how and to what extent the presence of the nano-sized filler alters the structure of the polymer matrix is of particular importance for obtaining materials with tailored properties. Even for the nanocomposites based on high-density polyethylene (HDPE), which are those that are most studied, the reported results are frequently contradictory or show no clear tendencies and relations. The types of nano-sized fillers applied so far for the preparation of HDPE-based composites include carbon nanomaterials (e.g., graphene oxide, reduced graphene, expanded graphite, glassy carbons, nanotubes [8,9,11,13,14,17,18,19]) particles of inorganic oxides (e.g., Al_2_O_3_ [20], MgO [21], TiO_2_ [10]), metals (e.g., Cu [22]), nano-clay [12] and organic nanoparticles [23,24]. These particles can have a different effect on the degree of crystallinity, including significant increase [12,15], decrease [17], and low- or no influence [10,25] on this parameter. Since crystallinity is the key factor responsible for a composite’s application characteristics, additional fundamental studies on the influence of the nanoparticles’ surface properties on the polymer matrix are required.

In this study, we investigated nanocomposite films prepared by filling the HDPE matrix with two fillers (ZrO_2_ and SiO_2_). Nano-SiO_2_ is a common polymeric additive that is successfully used as a filler in various fields [7,26]. Its polymer composites have excellent dielectric properties [27], high optical transparency [28], resistance to weathering [29] and abrasion [30]. Crystalline ZrO_2_ is known for its high chemical inertness and thermal stability. Nano-ZrO_2_ is a very promising material for creating radiation-resistant materials [31] with high mechanical and heat-shielding properties [32,33].

The problem of obtaining uniform dispersion of nano-fillers in a polymer matrix remains complicated. The quality of the dispersion as well as the size/shape and concentration of fillers are important for establishing the macroscopic properties of the resulting nanocomposite materials [34,35,36]. These effects have not yet been sufficiently studied.

The aim of this study is to partially fill the knowledge gap on the impact of metal oxide nano-fillers on an HDPE structure. The choice of ZrO_2_ and SiO_2_ as polymer fillers was dictated by their substantially different surface properties and adsorption affinity toward HDPE chains. This set of filler particles made it possible to partially elucidate the relationship between the surface properties of the filler and their impact on the structure and morphology of the HDPE matrix. As we show, the selected nanoparticles change the structure of the HDPE matrix in a completely different way. Based on this, we will classify nano-ZrO_2_ and nano-SiO_2_ as “inactive” and “active” and show that such an attribution is supported by the particle’s fractal characteristics.

## 2. Materials and Methods

### 2.1. Materials

Powdered high-density polyethylene (HDPE), identical to the one used in the work of [37], was used as a polymer matrix.

As fillers, we used two types of metal oxide nanoparticles: α-SiO_2_ and ZrO_2_ (Sky Spring Nanomaterials, Inc., Houston, TX, USA). The silica nanoparticles were amorphous [38] and 20~30 nm in diameter (specific surface area—S = 160 m^2^/g, density—2.65 g/cm^3^). The ZrO_2_ nanoparticles were crystalline (monoclinic phase) [38], with sizes in the range of 20~30 nm (specific surface area—S = 35 m^2^/g, density—5.68 g/cm^3^).

### 2.2. Nanocomposite Films Preparation

Polymer nanocomposite films of HDPE/α-SiO_2_ and HDPE/ZrO_2_ were prepared by the following steps. First, a mixture of HDPE with the desired amount of the nanoparticle filler (ZrO_2_ or SiO_2_) was prepared by melt-blending. Then, the mixture was subjected to hot pressing for 10 min (at 165 °C and 15 MPa). The final step was rapid quenching in the ice-water bath.

This approach, despite some drawbacks, like the limited mobility of polymer chains even at the molten state [36], is versatile and can be applied to a variety of thermoplastic polymers [39,40]. In addition, under carefully chosen conditions, it allows researchers to obtain composites with desirable properties. The volume fractions of nano-SiO_2_ and nano-ZrO_2_ in the composite were 1%, 3%, 5%, 10% and 20% *v*/*v*.

Pure HDPE films were used as a reference. The average diameter and thickness of nanocomposite films were ~5 cm and ~80–100 μm, respectively.

### 2.3. Small-Angle X-ray Scattering and Wide-Angle X-ray Scattering

Small-angle X-ray scattering (SAXS) and wide-angle X-ray scattering (WAXS) measurements were performed using two spectrometers. The SAXS patterns of HDPE composites with nano-SiO_2_ particles were obtained on a Rigaku X-ray instrument at MIPT, Dolgoprudny, Russia. An X-ray wavelength of λ = 1.54 Å was used, resulting in a momentum transfer *Q* in the range of (0.006–0.5) Å^−1^, where *Q* = (4π/λ)·sin (θ/2) and θ is the scattering angle [41].

The SAXS and WAXS measurements for HDPE and its composites with nano-ZrO_2_ particles were performed using a pinhole camera at the Institute of Macromolecular Chemistry CAS (Prague, Czech Republic). The momentum transfer (*Q*) range was 0.005—3.5 Å^−1^. And *Q* = (4π/λ)·sinθ, λ (in the given case 1.54 Å) is the wavelength and 2θ is the scattering angle [42,43,44].

### 2.4. Small-Angle Neutron Scattering

Small-angle neutron scattering (SANS) measurements were performed on the IBR-2 pulsed reactor on the YUMO spectrometer (JINR, Dubna, Russia) [45,46]. Experimental parameters: (1) the temperature range was 25–120 °C [47], (2) the distances from the sample to the detector were 5.28 and 13.04 m. The range of the scattering vector *Q* was (0.006–0.6) Å^−1^.

Neutron scattering intensities in absolute units (cm^−1^) were obtained by correcting the measured scattering spectra for transmission and sample thickness, background scattering on a film substrate, and on a reference vanadium sample using the small-angle scattering (SAS) software [48].

### 2.5. Transmission Electron Microscopy and Scanning Electron Microscopy

Transmission electron microscopy (TEM) analysis of the nanoparticles was performed on a Tecnai G2 F20 X-Twin high-resolution electron microscope (Eindhoven, Netherlands) with an operating voltage of 200 keV. The nanoparticles were dispersed in ethanol under pulsed low-power ultrasonication and then the suspension was dropped onto a formvar-coated TEM grid.

The cross-sectional fracture of nanocomposite films and the dispersion quality of the nanoparticles in the matrix were examined by scanning electron microscopy (SEM) using a JEOL JSM-6490LV microscope (Peabody, MA, USA) operating at 10 kV. To avoid deformations on the examined surface, the nanocomposite films were cryogenically fractured in liquid nitrogen.

### 2.6. Differential Scanning Calorimetry and Thermogravimetric Analysis

The thermal properties of composite films were analyzed using a NETZSCH 204 F1 Phoenix differential scanning calorimetry (DSC) Instrument (Selb, Germany). Samples of approximately 5–8 mg were placed in an aluminum crucible (with a diameter of 6 mm and a volume of 25/40 µL). For reference, an empty *Al* crucible was used. The DSC instrument was programmed to execute two consecutive heating/cooling cycles (from 25 to 200 °C and from 200 °C to 25 °C) for all samples. All measurements were performed in an argon atmosphere (70 mL/min) at a heating/cooling rate of 10 °C/min. Typical thermal parameters, e.g., enthalpy of fusion (Δ*H_m_*), enthalpy of crystallization (Δ*H_c_*), and the degree of supercooling (Δ*T* = *T_m_* − *T_c_*) were calculated for both the first and second cycle and were denoted with superscript “*” and “**”, respectively. The lamellar thickness was calculated from the Gibbs-Thomson equation [49,50]:
(1)$$l_{c}=\frac{2\sigma_{e}\cdot 10^{3}}{\Delta H_{m}\rho_{c}\left(1-\frac{T_{m}}{T_{m}^{o}}\right)},\mathrm{nm}$$
where $T_{m}^{o}$ is the theoretical melting point of polyethylene ($T_{m}^{o}$ = 145.7 °C) and *T_m_* is the experimental value; $\Delta H_{m}$ = 293 J × g^−1^, *σ_e_* is the lamellar surface free energy ($\sigma_{e}$ = 90 × 10^−3^ J × m^−2^, and $\rho_{c}$ is the density of crystalline polyethylene ($\rho_{c}$ = 0.94 kg × cm^−3^).

Thermogravimetric analysis (TGA) analysis was employed to determine the weight loss of the nanoparticle samples and to confirm the actual filler content in the composites. The TGA and differential thermogravimetric (DTG) curves were recorded using a TG 209 F1 Libra NETZSCH (Selb, Germany). About 10 mg of the sample was placed in an Al_2_O_3_ crucible (with 6.8 mm diameter and 85 μL volume). The nanopowder samples were heated under an argon flow (20 mL/min) in the thermobalance under dynamic conditions, over the temperature range of 30–900 °C at a heating rate of 10 °C/min. Here, the filler content was assumed as the residual mass, at 900 °C.

### 2.7. Vibrational Spectroscopy

Raman spectra were collected with 473 and 633 nm laser excitation sources on a Nanofinder 30 SOL Instruments spectrometer (Minsk, Belarus). Fourier transform-infrared (FT-IR) spectra were recorded using a single-reflection Smart iTR attenuated total reflection (ATR) accessory coupled to a Nicolet 6700 Thermo Scientific spectrophotometer (Lausanne, Switzerland). To minimize band shifts and intensity distortion related to the nature of ATR experiments, an advanced ATR correction algorithm implemented in OMNIC 9.2 software was used.

## 3. Results and Discussion

### 3.1. Characterization of the SiO_2_ and ZrO_2_ Nanoparticle Powders

First, we will discuss the small-angle scattering results for the nanoparticle powders. The combination of both SANS and SAXS, due to different scattering length density (SLD) values for X-rays and neutrons (Table 1), provides additional information on the size and dispersion of nanoparticles.

In the SANS and SAXS spectra of the pure nanoparticles, two power-law regimes, *Q*^−*D*^, can be distinguished (Figure 1a–d). Here, *D* denotes the negative value of the power-law exponent. According to Teixeira [51,52], the behavior of *I* (*Q*) at high *Q* values (*Q* > 0.02 Å^−1^) describes the surface of aggregates, whereas at *Q* < 0.02 Å^−1^ it characterizes their shape. The surface of the SiO_2_ clusters is described by *D*~3.8 (SAXS) and ~3.7 (SANS), characteristic of a rough surface. In the case of the ZrO_2_ nanoparticles, the *D* values of 4.08 and 4.12 (SAXS and SANS, respectively) are close to the Porod law (*Q*^−4^) describing scattering on a smooth aggregate surface (Table 2).

For small *Q* values (<0.02 Å^−1^), the X-ray scattering spectra are fitted, giving *D* the values of 3.08 and 2.2 for SiO_2_ and ZrO_2_ nanoparticles, respectively. In the case of neutron scattering, the respective *D* values were 2.65 and 2.12 (Table 2). In this *Q* range, the scattering power-law *Q*^−D^ with *D* < 3 is characteristic of the so-called mass fractals (ZrO_2_ case), whereas with 3 < D < 4 it describes surface fractals (SiO_2_ case).

Pair correlation functions P(r) for the X-ray and neutron scattering of nanoparticle powders obtained by the program GNOM [53,54] are shown in Figure 1e,f. If one considers the maximum at P(r) as the average distance between the particles and assumes that the particles are aggregated, the estimated mean size of ZrO_2_ nanoparticles is 13 nm for neutrons and X-rays. In the case of SiO_2_ nanoparticles, their mean size as deduced from neutron scattering (~15 nm) is much smaller than that obtained from X-ray scattering (~33 nm).

It is known that adsorbed water gives a sharp contrast to the surface of the aggregate, due to its negative scattering density in the case of neutrons; it effectively increases the size of the aggregates, due to the same effect of the scattering densities in the case of X-rays (see Table 1). This leads to a strong shift of the maxima on the P(r) curves obtained for the SiO_2_ sample from the SANS and SAXS data (Figure 1e).

Note also that because of the “smoothing” of the SiO_2_ particles’ surface due to adsorbed water, the Q exponents obtained from SANS are smaller than those from SAXS patterns.

Now, we will discuss the above experimental results. For ZrO_2_ nanoparticles, closely similar shapes of pair correlations functions and values of *D* exponents for SANS and SAXS measurements can be attributed to the fact that their surface is smooth, with only a low content of hydroxyl functional groups, and it adsorbs only negligible amounts of water. This is confirmed by supplementary TGA and FT-IR measurements, showing that the intensity of the hydroxyl (OH) stretching band at ~3500 cm^−1^ [55] is small, and the weight loss is only 0.8%, with a single peak in the differential thermogravimetric (DTG) curve at a temperature of 84 °C, suggesting the presence of weakly adsorbed water (Figure 2). The TEM image of the ZrO_2_ sample confirms that the particles are crystalline, and their surface is smooth. In addition, the particle size distribution, determined from TEM images, is in fairly good agreement with those figures deduced from the SAXS and SANS data (Figure 3).

In the case of the nano-SiO_2_ particles, the intensity of the hydroxyl-stretching band, due to the hydroxyl groups and adsorbed water, is much higher. This is reflected by a higher weight loss of ~6.3% and a more complex shape to the differential thermogravimetric (DTG) curve. Namely, a well-defined differential thermogravimetric (DTG) peak at ~98 °C, due to weakly adsorbed water, is accompanied by a long tail extending to temperatures as high as 650 °C. This highlights the presence of chemisorbed water and hydroxyl surface groups, which are known to decompose at ca. 400 °C [56]. Here, the TEM images show that the α-SiO_2_ particles are amorphous, of irregular shape, and with a highly rough surface. The average size of the particles is in the range of 25–35 nm (Figure 3). 

### 3.2. Small-Angle X-ray Scattering of Pure HDPE

The small-angle X-ray scattering from pure high-density polyethylene (HDPE) was well described by the paracrystalline model, in which the scattered intensity of a stack of layers is determined by:
(2)$$\begin{array}{c} I(Q)=N\cdot \Delta \rho^{2}\cdot P(Q)\cdot S(Q)\\ P(Q)=4\cdot \int_{0}^{\pi /2}{\left(\frac{J_{1}(QR\sin(\phi ))}{QR\sin(\phi )}\cdot \frac{\sin(QT\cos(\phi /2))}{QT\cos(\phi /2)}\right)}^{2}\sin(\phi )d\phi \\ S(Q)=1+\frac{2}{N}\cdot \sum_{k=1}^{N-1}(N-k)\cdot \cos(kQd)\cdot \exp\left(-k\frac{{\left(Q\sigma \right)}^{2}}{2}\right) \end{array}$$
where *N* is the number of layers forming the finite domain, *S* (*Q*) is the structure factor, *d* is the period distance, and *σ* is the parameter responsible for the degree of disorder of the second kind, i.e., where the long-range order gradually decreases with increasing *σ*. For simplicity, the form factor of one layer *P* (*Q*) is taken as a disk with the thickness as *T* and the radius, *R* [57,58].

The fitting of the experimental data using Equation (2), together with the form factor and the structure factor components, are shown in Figure 4.

The average cluster size in the pure polymer is about *N*_layer_ = 7 layers, with a thickness of the crystallized part of *l*_c_ = 8.6 nm, and a linear size of *L* ~2*R* = 15.4 nm. The thickness of the amorphous layer between the lamellae is about *l*_a_ = 10.2 nm. Taking into account the different densities of the amorphous (0.85 g/cm^3^) and crystalline phases (1.01 g/cm^3^), the estimated crystallinity degree of HDPE is ca. 50% and is consistent with DSC and WAXS results.

### 3.3. Effect of Filler Type and Concentration on the Crystallinity of the HDPE Matrix

In this article, we discuss the effect of adding both fillers on the crystal structure of the HDPE matrix. The WAXS patterns of HDPE and its composites with ZrO_2_ are presented in Figure 5. The WAXS pattern of pure HDPE shows strong (110) and (200) reflexes, which can be assigned to the orthorhombic structure [59].

We note that the (200) peak overlapped with (110), a reflection of the monoclinic ZrO_2_. The amorphous phase is represented by a broad peak at the *Q* range of ~1 to 2 Å^−1^. With the addition of ZrO_2_ filler, the parameters of the (110) diffraction peak, such as height, width, and position, practically do not change in the investigated range of filler content (0–20% *v*/*v*). In addition, the fraction of the polymer in the crystalline state remains relatively constant. This behavior qualitatively indicates that the lamellar size and the crystallinity degree of HDPE were almost unchanged for all filler concentrations.

This result is confirmed by DSC and Raman analysis. The DSC thermograms of HDPE and HDPE/%ZrO_2_ are presented in Figure 6a. Since the samples were rapidly quenched in an ice bath, their crystal structure may have differed markedly from the structure obtained by slow cooling. To analyze these differences, we performed two consecutive heating/cooling DSC cycles. The first heating cycle gives insight into the actual state of the sample, whereas the second one describes the state it achieves with a cooling rate of 10 °C/min. For a whole range of ZrO_2_ content, the peak melting temperatures (both *T_m_* * and *T_m_* **) did not vary significantly (Table 3).

The resulting lamellar thickness, calculated for the first cycle according to Equation (1), is in the range of 6.0–6.7 nm and is in agreement with the SAXS-based paracrystalline model (Equation (2)). Since the thickness of the lamellae in polyethylene is usually related to the degree of crystallinity, it can be expected that the degree of crystallinity will be almost unaffected by the addition of ZrO_2_ (up to 10% *v*/*v*). This is indirectly confirmed by the constant ratios of the enthalpies of fusion/crystallization, i.e., for all samples Δ*H_m_* */Δ*H_c_* * ~0.94 and Δ*H_m_* **/Δ*H_c_* ** ~1. Some increase in the lamellar thickness and degree of supercooling Δ*T* * is observed only at the highest ZrO_2_ loading of 20%. Direct information on the changes in crystallinity degree is provided by Raman spectroscopy. In the Raman spectra, the presence of the amorphous phase is manifested by several broad peaks at ~865, 1069, 1302, and 1443.5 cm^−1^ (overlapped with the σ (CH_2_) modes of the crystalline phase). The most prominent peaks, due to the crystalline phase, are at 1063, 1130, 1295, and 1416 cm^−1^. For nanocomposite films, the relative intensity of the signals due to the amorphous phase does not change with the addition of nano-ZrO_2_ [60].

Based on the above results, we classify the ZrO_2_ particles as “inactive” filler, i.e., virtually not influencing the crystalline structure of the HDPE matrix.

A radically different picture emerges for the nanocomposite films filled with silica particles. Here, *T_m_* ** is still constant but *T_m_* * gradually increases with increasing the nano-SiO_2_ content, and eventually *T_m_* * > *T_m_* **, at a filler content of 20% *v*/*v* (Table 3). Such behavior proves the thickening of the lamellae, from *l*_c_ = 6 nm (pure HDPE) to 8.3 nm (20% *v*/*v* SiO_2_).

On this basis, an increase in the crystallinity degree is expected to occur with the addition of SiO_2_ filler. In this case, we do not analyze the WAXS patterns since the signals of α-SiO_2_ and the amorphous polymer phase are strongly overlapped. However, the above assumption is directly evidenced by a gradual disappearance of the amorphous peaks in the Raman spectra (Figure 7b). Indirect evidence is also provided by comparing the relative changes in the enthalpy ratios: Δ*H_m_* */Δ*H_c_* ** and Δ*H_m_* */Δ*H_m_* **. Here, again, Δ*H_m_* **/Δ*H_c_* ** is constant and close to 1, whereas both Δ*H_m_* */Δ*H_c_* ** and Δ*H_m_* */Δ*H_m_* ** increase with an increase in the filler concentration.

In conclusion, SiO_2_ nanoparticles can be classified as “active”, i.e., capturing the crystallinity degree of HDPE and acting as a nucleation agent in rapid quenching conditions. This assignment/ascription is in agreement with previous studies showing the nucleation activity of silica particles on poly(ethylene terephthalate) [61] and polypropylene [62].

This behavior is explained within a thermodynamic approach, as used by Ebengou [63], assuming the nucleating activity to be an adsorption-driven process, and directly relating a reduction in the thermodynamic potential for nucleation with the interaction strength between the adsorbed polymer chains and the particles’ surfaces.

### 3.4. Small-Angle Neutron and X-ray Scattering of Polymer Nanocomposite Films

The SAXS and SANS complete scattering curves from HDPE/SiO_2_ and HDPE/ZrO_2_ in the 0–20% *v*/*v* at *t* = 25 °C are shown in Figure 8. In both cases (SAXS and SANS), the respective values of the *D* exponents for the composites are closely similar to those of pure nanopowders. This allows us to conclude that the surface roughness and fractal character of the particles do not change and are determined by their initial state in the powders. In addition, the scattering intensity contribution of the composites (1–20% *v*/*v*) scales fairly well with the filler mass fraction in the composite (φ).

Some differences and similarities between SANS and SAXS patterns can be explained, based on the SLD values (Table 1). Both SANS and SAXS can barely distinguish between the amorphous and crystalline polymer phases. In the case of composites, the contrast “seen” by SANS mainly comes from the differential SLD of the nano-filler and the polymer. Based on that finding, the main contribution to the total scattering is from the filler. In contrast, the scattering of the polymer matrix of the composite can be seen in SAXS experiments.

The addition of SiO_2_ nanoparticles to the polymer leads to the loss of correlation between the lamellae (Figure 8a). At 3% *v*/*v* HDPE/SiO_2_, there is still a weak correlation peak at the same value of the scattering vector *Q*. With a further increase in the composite concentration from 5% or higher, the correlations between the lamellae disappear. Schematically, the structures of pure HDPE and its composite with SiO_2_ particles are presented in Figure 9.

This result brings important new insights into the imperfections of the HDPE crystallization process in the presence of silica nanoparticles [64]. As is known, silica particles act as a nucleation agent, i.e., increasing the crystallization rate of HDPE. At the same time, SiO_2_ particles strongly adsorb polyethylene chains, thus limiting the mobility of the molecular segments. As a result, the rearranging of the segments requires additional energy [64]. In rapid cooling conditions, the rearrangement of HDPE segments seems to be suppressed and results in the disruption of lamellae ordering.

One noticeable phenomenon in SANS experiments is found by subtracting the scattering intensity of the nanoparticles, multiplied by the φ factor from the total scattering intensity of the corresponding composite. In the case of HDPE/SiO_2_, the resulting *I* (*Q*) is constant, reflecting the scattering of the polymer matrix, whereas in the case of HDPE/ZrO_2_, additional scattering appears. We attribute this additional scattering to the presence of voids, with a low density of polymer chains, that appear in the immediate proximity of ZrO_2_ nanoparticles. They are quite large and reach the order of ~10 nm (according to the spherical model fitting), which is comparable to the size of ZrO_2_ particles. It should be noted that the presence of voids was reported for some HDPE composites [16] and are assumed to be a result of poor adhesion between components and internal stresses induced in the matrix during quenching [16,65].

Such a difference in HDPE/SiO_2_ and HDPE/ZrO_2_ composites is in line with the above-presented conception of “active” (SiO_2_) and “inactive” (ZrO_2_) fillers. Particularly rough SiO_2_ particles act during crystallization as nucleation and adsorption centers.

Since the crystallization begins on the particles, rather than in the bulk, no cavity zones are present. Indeed, the SEM cross-sectional image of the HDPE/SiO_2_ composite shows that silica particles are built into the HDPE fibers and constitute an integral part of the nanocomposite (Figure 10a).

In the case of ZrO_2_ aggregates with an inert and smooth surface, the crystallization of the molten HDPE starts in the bulk polymer, and thus, the particles behave as a geometrical hindrance to the growing crystallites. Here, the cross-sectional SEM image of HDPE/ZrO_2_ is sharp, with clearly visible particle edges and holes resulting from detaching the particles and aggregates from the HDPE matrix upon the sample’s breaking (Figure 10b). This suggests that ZrO_2_ aggregates are not well adhered to the polymer matrix and that the integrity of the composite is poorer.

### 3.5. Thermal Stability of the Polymer Nanocomposite Structure

The effect of in-situ annealing on the structure of the nanocomposite films in the temperature range of 25 to 120 °C and filler contents of 3 to 20% (*v*/*v*) was investigated by SANS. Representative scattering curves for HDPE/SiO_2_ and HDPE/ZrO_2_ composite films, recorded at temperatures of 25, 80, and 120 °C, are shown in Figure 11.

For HDPE/SiO_2_ nanocomposite films, no structural changes were observed in the investigated range of temperatures and filler concentrations. Similarly, no changes for HDPE/ZrO_2_ at the lowest filler level (3% *v*/*v*) were detected on the size scale 1–100 nm (0.006–0.5 Ǻ^−1^) and temperatures 25–120 °C. Starting from a ZrO_2_ content of 5% (*v*/*v*), at a temperature of 80 °C, there is a transition in the SANS spectra. This observed change is related to the collapsing of the low-density polymer zones that surround the ZrO_2_ particles and is accompanied by a partial aggregation of the nanoparticles [66].

A rough estimation of the changes in the ZrO_2_ aggregates can be obtained from the Porod equation:
(3)$$I\left(Q\right)=2\pi \left(\rho_{\mathrm{nanoparticle}}-\rho_{\mathrm{polymer}}\right)\cdot Q^{-4}\cdot \left(S/V\right)$$
where *ρ*_nanoparticle_ and *ρ*_polymer_ are the neutron-scattering densities of nano-ZrO_2_ and high-density polyethylene (HDPE). *S*/*V* is the total area of the interface per unit volume of the ZrO_2_ aggregate. The specific surface area (*S*/*V*) of ZrO_2_ aggregates after heat treatment at 80 °C decreased by 3 and 10 times for filler concentrations of 5% and 20% *v*/*v*, respectively. For a threefold decrease in the *S*/*V* ratio, the predicted increase in the aggregate size is by a factor of 3 or 10, depending on the model used.

It should be mentioned that the structural transition is accomplished at a temperature of 80 °C, and a further increase in temperature up to 120 °C does not lead to any change in the size scale from 1 to 100 nm. This temperature is much lower than the melting point of the polymer matrix (130 °C). Evidently, the observed transition is due to the α-relaxation process. As known from NMR experiments, α-relaxation in polyethylene is related to helical jumps, i.e., 180° flip-flop motions accompanied by a displacement of the chain by one CH_2_ unit [67,68]. These motions occur even at room temperature (~10 jumps/s), but a considerable jump rate of ~10^4^/s is attained only at a temperature of 360 K (~87 °C). At this temperature, the chain displacement is several nm per second and can occur between amorphous and crystalline regions. Such a picture is in good relationship with the transition temperature range at SANS measurements and the dynamic of α-relaxation of the neat HDPE sample from temperature-dependent dielectric measurements [69,70]. These transformations in the nanoscale level give a new insight into the changes in mechanical properties (e.g., tensile strength [71]) observed in some quenched composites after mild annealing.

Since the silica particles act as the adsorbing and nucleating agents of HDPE crystallization, the low-density regions in the vicinity of the nanoparticles are absent and, thus, no transition is observed in the SANS spectra.

## 4. Conclusions

We studied the interfacial effects in HDPE-nanoparticle composite films, prepared using two different filler particles: nano-ZrO_2_ and nano-SiO_2_.

For both fillers, we found that particle nanoaggregates were evenly distributed in the polymer matrix, and their surface roughness and fractal character were determined by their initial state in the powders.

We demonstrated that the surface properties of the nanoparticles have a primary impact on the structure of the polymer matrix. SAXS and SANS data, supported by TEM imaging, revealed that ZrO_2_ particles show mass fractal characteristics (*D* ~4.08, SAXS and 4.12, SANS) with inert and smooth surfaces, whereas the SiO_2_ particles were surface fractals (*D* ~3.8, SAXS and ~3.7, SANS) of irregular shape and developed surface area. These properties allowed us to categorize the particles as “inactive” (ZrO_2_) and “active” (SiO_2_) and describe their fundamentally different effects on the HDPE matrix.

In the case of nano-ZrO_2_ filler, the lamellar thickness and crystallinity degree remain unchanged over a wide range of filler concentrations. We propose that, in the presence of nano-ZrO_2_ filler, the crystallization of HDPE starts in the body rather than at the surface. Here, the nanoparticles act as a geometrical hindering factor for the growing lamellae. As a result, the particles have poorly adhered to the matrix, and zones of reduced polymer density (i.e., voids of ~10 nm in size) surrounding the particles are formed. As evidenced by the temperature-dependent SANS measurements, these zones collapse at a temperature of 80 °C with the simultaneous partial aggregation of the nanoparticles. The phenomenon is driven by polymer chain displacements related to the α-relaxation process.

Silica nanoparticles exert a completely different impact on the crystallization of HDPE under rapid quenching conditions. We found that both the lamellar thickness and the degree of crystallinity increase with an increase in the filler loading. Based on this finding, we conclude that nano-SiO_2_ particles are “active” in HDPE crystallization, acting as nucleation centers for lamellae growth. Since the crystallization starts on the particles’ surface, not in the body, no cavity zones are present, and also no changes in temperature-programmed SANS experiments are visible. Based on SAXS measurements, we found that the ordering of the lamellae is disrupted even at a filler content of a few percent. This result provides new insight into peculiarities in HDPE/SiO_2_ crystallization related to the limited mobility of the molecular segments adsorbed on the particles.

The present results allow a better understanding of the changes in the polymer matrix in a presence of nanoparticles of different surface properties and can be used in composite design. The different behavior of both fillers is expected to lead to different mechanical and dielectric properties in the nanocomposites, and this will be the subject of our further studies.

## Figures and Tables

**Figure 1 nanomaterials-11-02673-f001:**
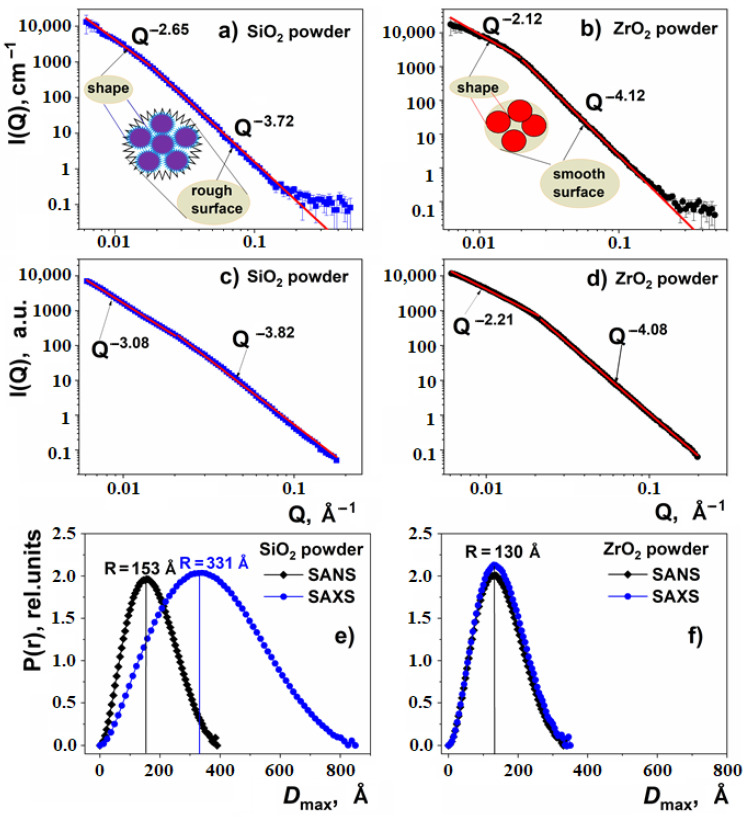
Small-angle scattering data for SiO_2_ (left column) and ZrO_2_ (right column): (**a**,**b**) SANS spectra, (**c**,**d**) SAXS spectra, (**e**,**f**) pair correlation functions for SiO_2_ and ZrO_2_ nano powders for SAXS and SANS scatterings.

**Figure 2 nanomaterials-11-02673-f002:**
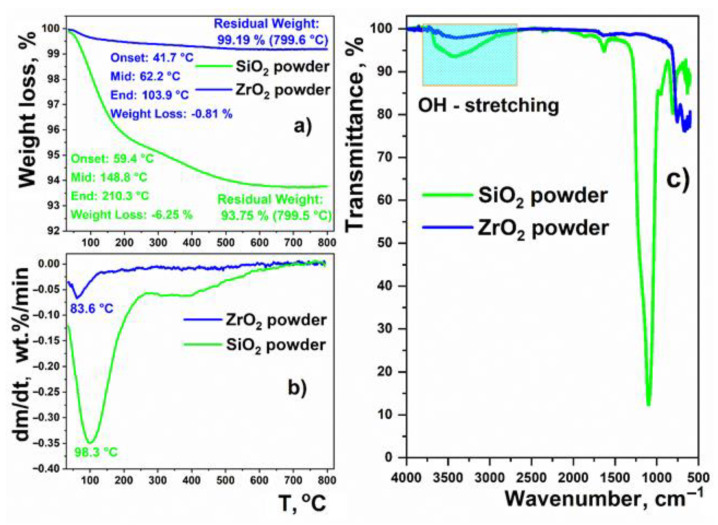
TGA, DTG (**a**,**b**) curves and FT-IR spectra in the mid-infrared range (**c**) of nano-SiO_2_ and nano-ZrO_2_ powders.

**Figure 3 nanomaterials-11-02673-f003:**
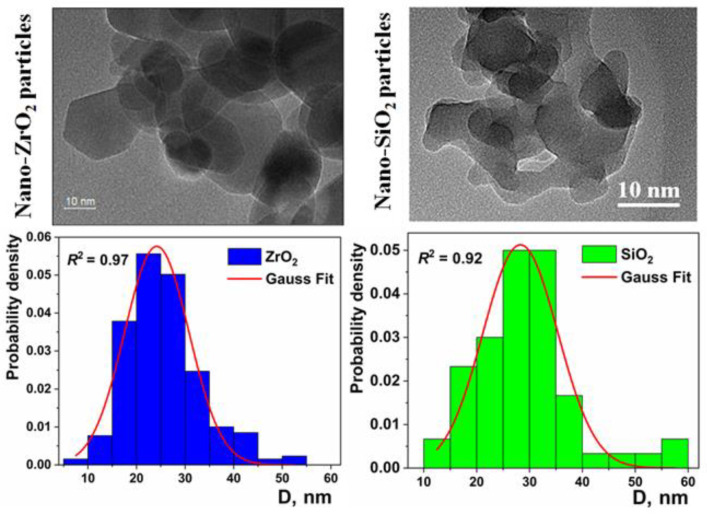
TEM images and normalized particle size distributions for pure nanoparticles: ZrO_2_ (**left column**) and SiO_2_ (**right column**).

**Figure 4 nanomaterials-11-02673-f004:**
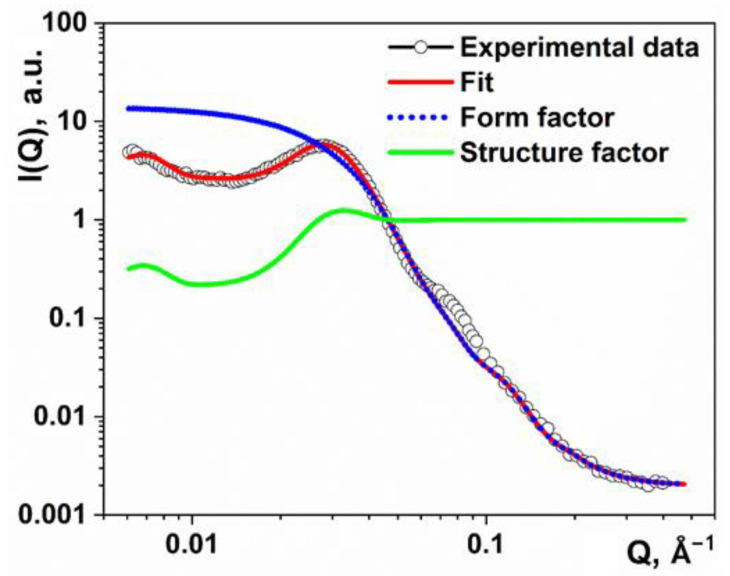
SAXS curves for the pure high-density polyethylene (HDPE).

**Figure 5 nanomaterials-11-02673-f005:**
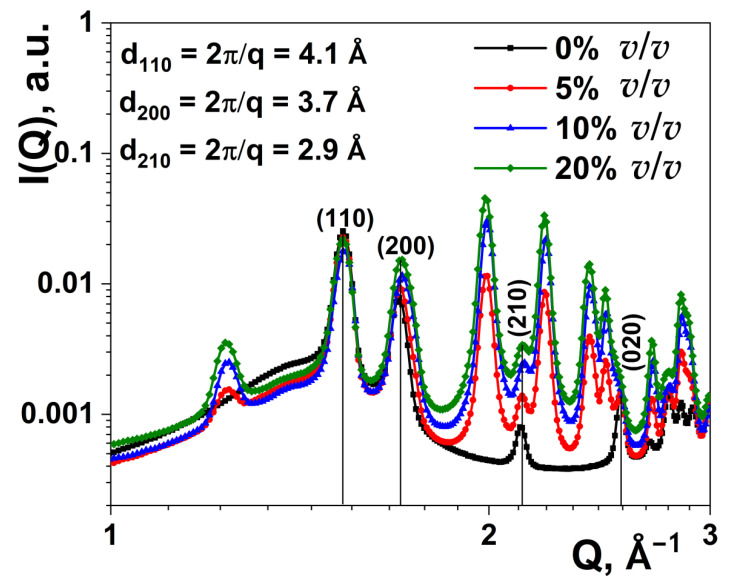
Wide-angle X-ray scattering patterns of HDPE and its composites with ZrO_2_ nanoparticles.

**Figure 6 nanomaterials-11-02673-f006:**
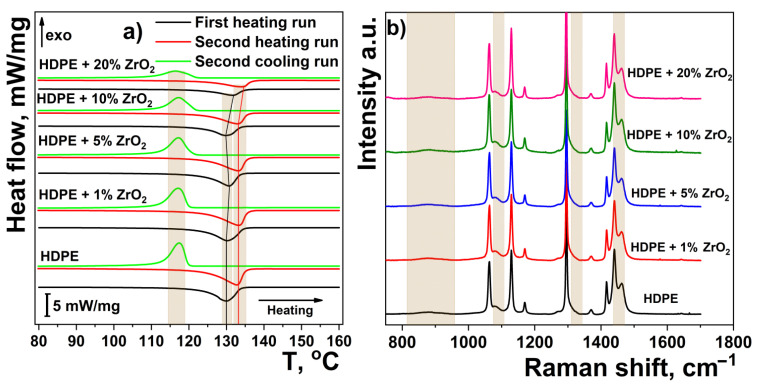
DSC thermograms (**a**) and Raman spectra (**b**) of HDPE and HDPE/%ZrO_2_ composite.

**Figure 7 nanomaterials-11-02673-f007:**
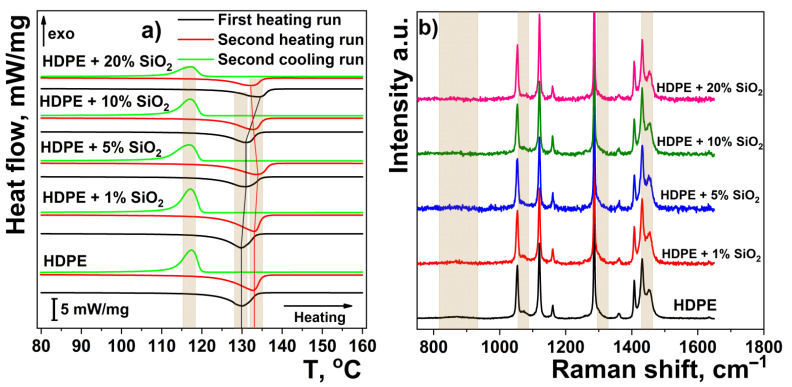
DSC thermograms (**a**) and Raman spectra (**b**) of HDPE and its composites with nano-SiO_2_ particles.

**Figure 8 nanomaterials-11-02673-f008:**
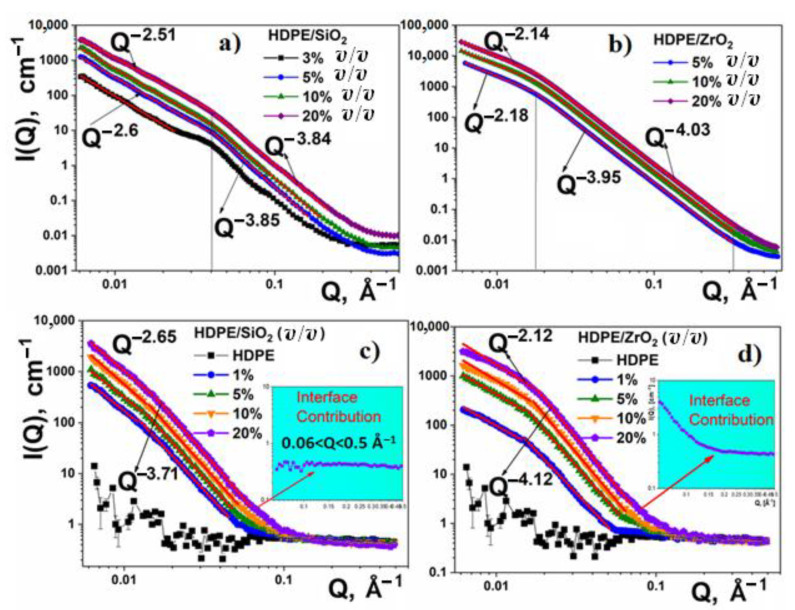
SAXS (**a**,**b**) and SANS (**c**,**d**) scattering spectra of the HDPE/SiO_2_ (**left column**) and HDPE/ZrO_2_ (**right column**) composites. Insets show additional interface scattering after subtracting the scattering intensity of the nanoparticles multiplied by the φ factor from the total scattering intensity of the corresponding composite.

**Figure 9 nanomaterials-11-02673-f009:**
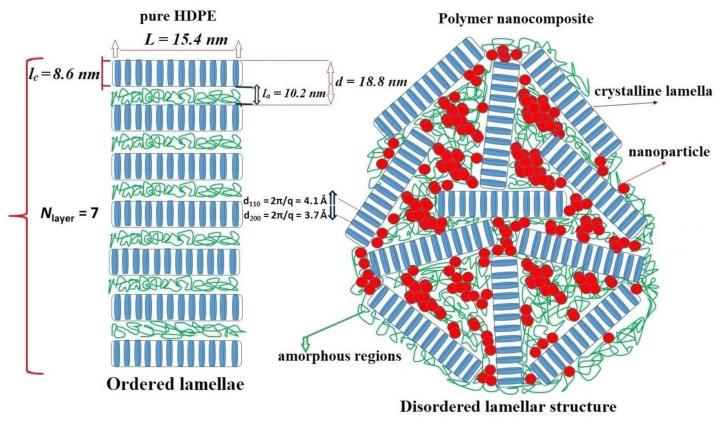
Schematic illustration of the impact of SiO_2_ nanoparticles on the lamellar structure of the high-density polyethylene (HDPE) matrix.

**Figure 10 nanomaterials-11-02673-f010:**
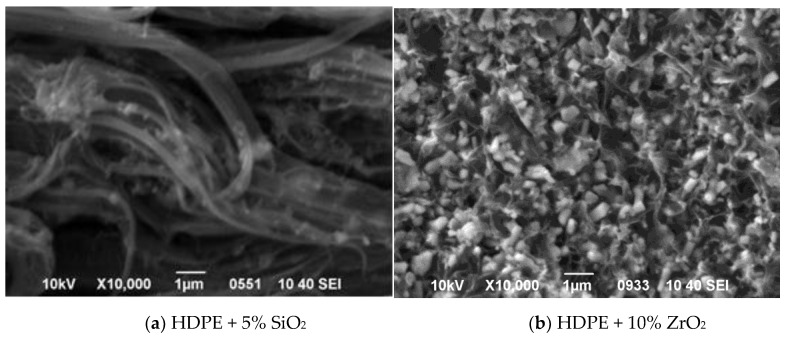
Cross-sectional SEM images of HDPE/SiO_2_ (**a**) and HDPE/ZrO_2_ (**b**) composite films, with filler loading of 5% and 10% (*v*/*v*), respectively.

**Figure 11 nanomaterials-11-02673-f011:**
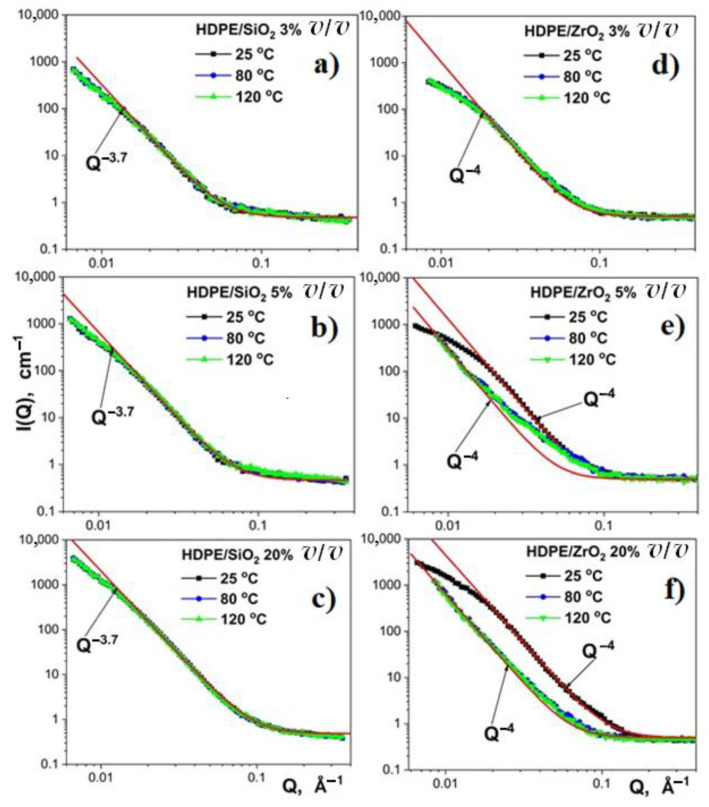
Small-angle neutron scattering from: (**a**) HDPE + 3% SiO_2_; (**b**) HDPE + 5% SiO_2_; (**c**) HDPE + 20% SiO_2_; (**d**) HDPE + 3% ZrO_2_; (**e**) HDPE + 5% ZrO_2_ and (**f**) HDPE + 20% ZrO_2_ composite films at different temperatures.

**Table 1 nanomaterials-11-02673-t001:** X-ray and neutron-scattering length densities (SLD) for polymer/filler complex.

Sample	Density (g/cm^3^)	SLD (Neutron) (cm^−2^)	X-ray (cm^−2^)
ZrO_2_	5.68	5.21 × 10^10^	43.1 × 10^10^
SiO_2_	2.65	4.19 × 10^10^	22.6 × 10^10^
Polyethylene (semi-crystalline)	0.95	−0.340 × 10^10^	9.21 × 10^10^
Polyethylene (crystalline)	1.01	−0.361 × 10^10^	9.79 × 10^10^
Polyethylene (amorphous)	0.85	−0.304 × 10^10^	8.24 × 10^10^
H_2_O	1.0	−0.561 × 10^10^	9.44 × 10^10^

**Table 2 nanomaterials-11-02673-t002:** Characteristic values of the *D* exponent in the scattering power-law *Q*^−D^ for nanoparticle powders.

Nanoparticle	SAXS		SANS	
	Q < 0.02 Å^−1^	Q > 0.02Å^−1^	Q < 0.02Å^−1^	Q > 0.02Å^−1^
**SiO_2_**	3.08	3.82	2.65	3.72
**ZrO_2_**	2.21	4.08	2.12	4.12

**Table 3 nanomaterials-11-02673-t003:** The DSC melting and crystallization characteristics for pure HDPE and its nanocomposites. Here: ω—filler content, *T_c_*—crystallization temperature, *T_m_*—melting temperature, Δ*H_m_*—enthalpy of melting, Δ*H_c_*—enthalpy of crystallization, *l*_c_—lamellar crystal thickness, and Δ*T*—degree of supercooling.

ω, vol. %	ω mass.%	*T_m_* * °C	*T_m_* ** °C	*T_c_* * °C	*T_c_* ** °C	Δ*H_m_* * J·g^−1^	Δ*H_m_* ** J·g^−1^	*l*_c_ * nm	*l*_c_ ** nm	Δ*H_c_* * J·g^−1^	Δ*H_c_* ** J·g^−1^	Δ*T* * °C	Δ*T* ** °C
**HDPE + %SiO_2_**													
0	0.0	129.8	132.8	117.4	117.3	−170.5	−203.2	5.9	7.3	187.9	184.3	12.5	15.5
1	2.7	129.8	133.0	117.2	117.2	−170.0	−204.8	5.9	7.5	196.3	197.6	12.6	15.8
5	12.7	130.7	133.6	116.8	116.8	−142.4	−171.6	6.4	7.9	169.4	171.0	13.9	16.8
10	23.5	131.1	132.8	117.1	117.1	−132.1	−150.0	6.4	7.4	153.7	154.1	14.0	15.7
20	40.8	134.2	132.4	117.2	117.3	−115.9	−107.7	8.2	7.1	106.2	106.2	16.9	15.2
**HDPE + %ZrO_2_**													
0	0.0	129.8	132.7	117.4	117.3	−170.5	−203.2	5.9	7.3	187.9	184.3	12.4	15.4
1	5.7	130.2	133.2	117.2	117.1	−165.6	−187.4	6.1	7.7	176.6	167.2	13.0	16.2
5	23.8	130.7	133.2	117.2	117.2	−151.5	−174.0	6.3	7.6	161.2	157.6	13.5	16.0
10	39.7	130.1	133.0	117.2	117.3	−119.6	−146.2	5.9	7.5	132.4	131.8	12.9	15.7
20	59.7	131.7	134.1	116.3	116.5	−82.4	−96.3	6.7	8.1	91.78	90.3	15.4	17.5

*—the values obtained from the 1st heating and cooling cycles; **—the values obtained from the 2nd heating and cooling cycles.

## Data Availability

The data that support the findings of this study are available from the corresponding author: A.A.N., upon reasonable request.

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
