# Peer review of "Composite Films of HDPE with SiO2 and ZrO2 Nanoparticles: The Structure and Interfacial Effects"

_nanomaterials, 2021, doi:10.3390/nano11102673_

Round 1
Reviewer 1 Report
This paper presents the study of the interfacial behavior of two different filler particles, nano-ZrO2 and nano-SiO2. Small-angle X-ray scattering (SAXS), wide-angle X-ray scattering (WAXS), small-angle neutron scattering (SANS), transmission electron microscopy (TEM), scanning electron microscopy (SEM), differential scanning calorimetry (DSC), thermogravimetric analysis (TGA), and vibrational spectroscopy have been implemented to characterize their behavior in the matrix of high-density polyethylene (HDPE).
While the experiments involved significant amount of work, the paper in its current form does not provide a clear motivation behind of such an effort. I don’t mean you had no motivations or contributions, but it is at lease not clear to the readers in the current form of the paper. It is not explicitly clear what the motivation for this study is.
In addition, the paper does not have enough rigors in analysis and discussion for publishing in nanomaterials. It reads like a technical report describing the experimental observations. Although the authors cited several references to explain their observations, but it is not enough to assess or support their theory. The authors should make an effort to include more experimental evidence or analytical analysis to support what is written in the manuscript.
The abstract should state the novelty, should be well organized, well defined, and concise. The original aspects and contributions of the study should be emphasized in abstract which should also clearly indicate the potential uses of the results, and the implications of the proposed method.
In the conclusion section, important new results and knowledge along with their potential use should be listed as quantitative as possible. It should not just summarize what work was conducted and observed in the manuscript.
Lastly, although the English can be easily followed it is must be further improved throughout. Moreover, an acronym should be written out in full the first time it is used on a page, followed by the abbreviation in brackets.
Reviewer 2 Report
Comments: This manuscript reported the investigation on structural properties of two different kinds of nanocomposite films, namely the HDPE/nano-SiO2 and HDPE/nano-ZrO2 films with a variety of measurement methods, SAXS, SANS, WAXS, DSC, TGA, TEM, SEM, Raman spectra, and FT-IR spectra. The crystallinity, lamellae thickness, and the evolution of the structure of the composite at heating temperature of HDPE is studied. Unfortunately, the manuscript is not well arranged, the level of novelty is not strong enough to warrant its publication in this Journal.
- The main subject of the manuscript is confusing. It seems that the manuscript reported how to use these measurement methods to characterize the structure of the composite rather than the structure of the HDPE composite. Many of the structure information of the composite are not shown, such as the lamellae thickness, the crystallinity, etc.
- The application of measurement methods is puzzling.
For example, the authors charactered the surface and size of nano-SiO2 and nano-ZrO2 with SAXS and SANS, it’s hard to understand why they don’t use SEM, TEM, and particle size analyzer to obtain these information, which are much more intuitive and accurate.
In addition, there is no detailed information about how the SAXS and SAXS measurements on the particle size and morphology were carried out, e.g., in dispersion or in HDPE composite?
- The lamellae thickness of the composite was calculate with DSC measurement. The reviewer wonder why the SAXS result is not used to calculate the lamellae thickness or long period of HDPE. Theoretically, the melting point of polymer depends on the lamellar thickness of HDPE, but few researcher would used DSC rather than SAXS result to calculate the lamellar thickness.
- The crystallinity is an important information about the composite, which should be calculated with the WAXS result. It’s not easy to get the crystallinity with the integrated one-dimensional plots in Figure 5.
- The novelty of this work are suggested to be emphasized.
Reviewer 3 Report
- There are so many polymers, why HDPE was chosen as the research object (polymer matrix)? As stated by the authors: “The quality of the dispersion, as well as the size/shape, and concentration of fillers are important for the macroscopic properties of the resulting nanocomposite materials [16-18].”, my question is there are so many metal oxide nanoparticles, why two kinds of nanoparticles of SiO2 and ZrO2 were used as nanofillers? Why nanoparticles rather than nanofibers, nanosheets or nanobelts were used? The related illustrations and discussions should be added in the Introduction part of manuscript.
- It is not attractive enough in terms of material design and preparation. Up to now, many similar works have been published, the authors should highlight some points in the system design and research methods.
- In the manuscript, the authors discussed the effect of adding both fillers on the crystal structure of the HDPE and posted the conception of ‘active’ (SiO2) and ‘inactive’ (ZrO2) filler. However, from the molecular perspective, the micro combination mechanism between matrix and fillers should be further investigated and discussed.
- TEM images shown in Fig. 3 indicate that the nanoparticles were severely agglomerated. Have the two nanoparticles been used directly without further processing? Besides, low-magnification TEM images cannot prove whether the sample crystallizes and cannot be used to characterize the smoothness of the sample surface. It is recommended to provide high-magnification SEM or TEM characterization.
- Check that whether all images are processed as required or not. Such as, add the labels (a, b, c, d …) to each picture in Fig. 3 to correspond to the description in the manuscript.
- Enlarge the amorphous peaks of the Raman spectra for comparison. The current spectra in Fig.7 b show no significant difference.
- “The defocused images”, “visual inspection”, etc., should provide relevant evidence to confirm, rather than simple expression.
- Although the authors have studied whether the "active" (SiO2) and "inactive" (ZrO2) fillers act during crystallization as nucleation centers, which is a key factor in improving the crystallinity of HDPE. However, in the first half of the paper, a lot of space was devoted to characterizing the surface roughness, density, size, dispersion state of nanoparticles, and the differences between surface hydroxyl groups and adsorbed water, and then the mechanism of the effect of this series of parameters on the crystallization performance of HDPE are worth further discussion.
- English should be greatly improved, there are many grammar mistakes/typo errors in the manuscript and the authors should go through the manuscript carefully. Specially, the abbreviations of HDPE, SANS, SAXS, and DTG, etc. should be defined when they appeared in the text for the first time and then be used throughout the text.
Reviewer 4 Report
In this manuscript the authors reported the investigation of two different kinds of nanocomposite films. Some interesting results are obtained. The lamellar thickness and the degree of crystallinity increase with increasing the nano-SiO2 filler loading. I therefore recommend an acceptance for publishing after next revisions.
1.Pages 1, abstract part, some background sentences can be added;
2.Introduction part, if possible, some important and relative applied reports about self-assembled film nanostructures from various styles (Optics Communications, 2021, 481: 126522.; Opt. Express, 2021, 29(4): 5152-5165.; ACS Omega, 2021, 6(7): 4958-4967.; Chemical Engineering Journal, 2021, 417: 129233.) should be added to show clear background;
3. how about the BET and roughness for 2 kind of films?
4. what about the stability for nanocomposite films, pleas add more describe?
5. Some minor Language error and style should be modified.
Round 2
Reviewer 2 Report
The authors has already revised the manuscript according to the review comments.